# Maintaining Proportional Committees with Dynamic Candidate Sets

**Chris Dong** [1]  **Jannik Peters** [2]

## Abstract

Multiwinner voting is the study of electing a fixed-size committee given individual agents' preferences over candidates. Most research in this field has been limited to a static setting, with only one election over a fixed set of candidates. However, this approach overlooks the dynamic nature of applications, where candidate sets are subject to change. We extend the study of proportionality in multiwinner voting to dynamic settings, allowing candidates to join or leave the election and demanding that each chosen committee satisfies proportionality without differing too much from the previously selected committee. We consider approval preferences, ranked preferences, and the proportional clustering setting. In these settings, we either provide algorithms making few changes or show that such algorithms cannot exist for various proportionality axioms. In particular, we show that such algorithms cannot exist for ranked preferences and provide amortized and exact algorithms for several proportionality notions in the other two settings.

## 1. Dynamic Committee Selection

Given diverse preferences of agents, a central subject of research in artificial intelligence, collective decision-making, and social choice theory is the task of choosing a subset of possible candidate items that proportionally represents the agents' preferences. Applications include clustering (Caragiannis et al., 2024; Kellerhals and Peters, 2024), facility location (Jung et al., 2020), query answering, and top-$k$ selection protocols (Behar and Cohen, 2022; Islam et al., 2024), or democratic innovations like participatory budgeting (Peters et al., 2021) or civic participation platforms (Fish et al., 2024). Most of the previously mentioned papers derive their models from the basic social choice task

of multiwinner voting: $n$ voters submit preferences over a subset of candidates, from which $k$ need to be selected. For instance, in the classical centroid clustering problem, the voters correspond to the points to be clustered, the candidates are the possible cluster centers, $k$ is the number of cluster centers to be selected, and the preferences are derived from the distances between the points and the cluster centers.

In the vast majority of settings studied so far, multiwinner voting is "one-shot": voters and candidates are given, and the task is to make one selection only. However, assuming such a static setting might be too restrictive for real-world applications (a concern raised, for instance, by Boehmer and Niedermeier, 2021; Elkind et al., 2024). In election settings, candidates might concede their seat and then need to be replaced (Gallagher, 1996); in hiring settings, candidates could apply for the job or decline offers at any time during a hiring procedure (Brill et al., 2023); and in facility location problems, new possible locations might open up.

As a possible application, consider the setting of civic participation platforms (Halpern et al., 2023; Fish et al., 2024). A website (such as https://pol.is, studied by Halpern et al.) wants to curate a slate of representative *new* comments on its homepage. In this setting, the comments correspond to candidates, and the voters correspond to the users of the website. The voters express which comments they like, for instance, by up- or down-voting them. Such a setting, however, is inherently "online", as new comments get posted, and old comments get removed when they become out-of-date. Our work studies if it is possible to maintain a slate of representative comments, while only making a small amount of changes with each update.

We follow in the footsteps of recent work in the matching (Matuschke et al., 2019; Bernstein et al., 2019), and clustering (Bhattacharya et al., 2024; Łącki et al., 2024) literature and study dynamic multiwinner voting. In particular, we study *low recourse* dynamic proportional multiwinner voting and answer the following question: can we select proportional committees such that after the candidate set changes, we can restore proportionality by altering the original committee slightly?

In a recent paper Chen, Hatschka, and Simola (2024) examined a highly related problem: given two different commit-

---

*Equal contribution [1]TU Munich [2]National University of Singapore. Correspondence to: Chris Dong <chris.dong@tum.de>, Jannik Peters <peters@nus.edu.sg>.

*Proceedings of the 42$^{nd}$ International Conference on Machine Learning*, Vancouver, Canada. PMLR 267, 2025. Copyright 2025 by the author(s).

tees, can we transform one of the committees into the other by swapping candidates in and out of the committee. However, their study focuses only on committees elected by specific scoring rules in an approval-based setting, and provides results exclusively limited to computational intractability and fixed parameter tractability. Crucially, while some scoring rules always select proportional committees, Chen et al. (2024) only require that the in-between committees have bounded score differences. However, simply bounding the differences in scores does not guarantee that the committees selected in-between are actually proportional. In contrast, our work focuses on guaranteeing proportionality itself.

**Our Results** We study the dynamic preservation of proportionality in three common settings, where preferences are modeled via linear orders, approval ballots, and distances in a metric space. We further distinguish between the three classical dynamic frameworks where the candidate set can iteratively grow, decrease, or both. For linear orders, we show that the popular axiom of *proportionality for solid coalitions (PSC)* (Dummett, 1984) can be maintained in the incremental framework, but not in the decremental or fully-dynamic framework, where the number of required replacements after a single deletion may be arbitrarily large. For approval ballots, we provide fully-dynamic algorithms that satisfy the notions of PJR+ and $\mathcal{O}(\log(k))$-EJR+ (Brill and Peters, 2023) while requiring, on average, two changes per time-step. Additionally, we show that in the incremental setting $\alpha$-EJR+ can be satisfied for constant $\alpha > 1$ with the average number of swaps per time-step being constant and independent of $k$. For proportional clustering, we obtain a fully-dynamic algorithm that achieves a 4.24-proportional fair outcome and a 5 approximation to the $q$-core (Kellerhals and Peters, 2024; Ebadian and Micha, 2025). Surprisingly, this meets the *currently best known* upper bound on the $q$-core of non-dynamic algorithms. Hence, we achieve results that are more positive than one might anticipate based on Chen et al. (2024), especially in proportional clustering and for the incremental setting.

**Further Related Work** Proportionality in the static multiwinner voting setting is well studied. In detail, we refer to Lackner and Skowron (2023) for a recent book on approval-based multiwinner voting and to the works of Aziz and Lee (2020; 2021; 2022), Brill and Peters (2023), and Bardal et al. (2025) for proportionality with ranked preferences. Further, we refer to Chen et al. (2019) for the introduction of proportional fairness in clustering; to (Micha and Shah, 2020; Li et al., 2021; Kalayci et al., 2024) for follow-up work; and to (Aziz et al., 2024; Kellerhals and Peters, 2024) for relating it to multiwinner voting and the problem of individual fair clustering (Jung et al., 2020; Mahabadi and Vakilian, 2020).

As for other online models of multiwinner voting, our work

is related to Do et al. (2022). In their approval-based setting, candidates appear in an online manner and have to be irrevocably chosen or rejected at each time-step. The difference is that in our model (i) the committee needs to satisfy the proportionality notion at every step of the online process and (ii) chosen candidates can be taken off the committee or even off the feasible set again. Further, Brill et al. (2023) consider a model in which they assume that the set of candidates is known in advance, but the actual availability of the candidates is unclear and needs to be requested via an invitation to the committee. Similarly to Do et al. (2022), candidates are added irrevocably. Further, the preferences over all candidates are known in advance. In a series of works Deltl et al. (2023); Bredereck et al. (2022; 2020) considered the complexity of the *sequential committee selection problem*. In their model, the rules select one committee per time-step, while constantly satisfying requirements such as optimizing egalitarian welfare or scores, subject to the committees not changing drastically over time. Most problems studied in these works are computationally intractable and thus the authors instead study the parameterized complexity of several related problems.

## 2. Model and Notation

In this section, we first introduce a general static framework, then expand it to be dynamic. Setting-specific definitions are in the corresponding sections.

**Static Framework** Throughout the paper, we denote by $N = [n]$ the set of voters and by $C^* = \{c_i : i \in \mathbb{N}\}$ the set of candidates. A *preference profile $P$* contains the voters' preferences over the candidates in $C$. Further, when given some $C \subseteq C^*$, we denote the *restriction* of $P$ to $C$ by $P|_C$. The exact form of $P$ and $P|_C$ depends on the setting and we provide according definitions in each corresponding section. Roughly speaking,

(i) with *ordinal preferences*, each voter $i \in N$ has a linear order over $C^*$;

(ii) with *proportional clustering*, preferences are given via distances in a pseudometric space $(N \cup C^*, d)$; and

(iii) with *approval preferences*, each voter $i \in N$ has an *approval set* $A_i \subseteq C^*$.

Given a *target size $k$*, a profile $P$, and a finite set $C \subseteq C^*$ of size at least $k$, the goal is to select a committee $W \subseteq C$ of size $k$. For each $k \in \mathbb{N}$, $C \subseteq C^*$ with $|C| \geq k$, and preference profile $P$, the tuple $\mathcal{I} = (C, P|_C, k)$ constitutes an (approval/ordinal/clustering) *instance*. A *static voting rule $f$* takes instances $\mathcal{I}$ as input and outputs a committee $f(C, P|_C, k) \subseteq C$ of size $k$. Throughout the paper we write that a group $N' \subseteq N$ of voters is $\ell$-*large*, if $|N'| \geq \frac{\ell n}{k}$.

Intuitively, for an outcome to be proportional, every $\ell$-large group of voters should be represented with $\ell$ candidates in the outcome.

**Online Algorithms**  We introduce three settings for dynamic algorithms. In each of these, a sequence of instances with varying candidate sets is given, and the algorithm is only allowed to use information up to the current time-step.

First, in the *incremental setting*, we are given a set $C_0$ of $m \geq k$ candidates together with a stream of candidates $c_1, c_2, \ldots$, appearing over time. Accordingly, the voters' preferences are revealed incrementally. For this purpose, let $t \in \mathbb{N}$ denote a *time-step* and $C_t = C_0 \cup \{c_1, \ldots, c_t\}$ be the set of available candidates at this time-step. Formally, an *incremental algorithm* $f$ takes as input $(C_t, P|_{C_t}, k, t)$ and outputs a committee $f(C_t, P|_{C_t}, k, t) \subseteq C_t$ of target size $k$.

Next, in the *decremental setting*, we start with some set of candidates $C_0 = \{c_1, \ldots, c_m\}$, $m \geq k$, and the voters reveal their preferences over $C_0$ immediately. However, the order in which candidates drop out of the election is not known to $f$ in advance. Formally, any sequence $(C_t)_{t=0,\ldots,m-k}$, with $|C_t| = m-t$ and $C_t \subset C_{t-1}$ is admissible and a decremental algorithm $f$ takes as input instances of the form $((C_s)_{s \leq t}, P|_{C_0}, k, t)$ and outputs a committee $f(C_t, P|_{C_0}, k, t) \subseteq C_t$ of target size $k$.

In the *fully-dynamic setting*, the candidates can both join and leave. As in the decremental setting, we start with $C_0 = \{c_1, \ldots, c_m\}$ for some $m \geq k$. However, the only requirement on the sequence $(C_t)_{t \geq 0}$, apart from $|C_t| \geq k$, is that $C_{t+1}$ is obtained from $C_t$ by adding or removing one candidate for all $t \in \mathbb{N}$. A fully-dynamic algorithm takes as input $((C_s)_{s \leq t}, P|_{\bigcup_{s \leq t} C_s}, k, t)$ and outputs a committee $f((C_s)_{s \leq t}, P|_{\bigcup_{s \leq t} C_s}, k, t) \subseteq C_t$ of target size $k$.

Let a (fully-dynamic/decremental/incremental) algorithm $f$ be given. Once a sequence $(C_t)_t$ is fixed, the complete input of $f$ can be derived from the tuple $(P, k, t)$, and for brevity we hence write $f(P, k, t)$ in all three frameworks. Note that, despite this shorthand notation, $f$ can only use information that is available up to time-step $t$ and has finite input. Often, even $P$ and $k$ will be clear from the context, in which case we abbreviate $f(t)$.

Let $X$ be any proportionality axiom. An algorithm $f$ satisfies $X$ if $f(P, k, t)$ satisfies $X$ for all $k$, $t$, and $P$. A (fully-dyamic/decremental/incremental) algorithm $f$ is *robust*, if for all (fully-dyamic/decremental/incremental) instances we have $|f(P, k, t) \cap f(P, k, t+1)| \geq k - 1$. Finally, we say that an algorithm makes *amortized* $\ell$ changes if after $t$ rounds $\sum_{t'=0}^{t-1}(k - |f(P, k, t') \cap f(P, k, t'+1)|) \leq t \cdot \ell$.

Throughout the paper, we will reference well-known rules from the static multiwinner voting setting to illustrate that the dynamic problems add a layer of depth that has yet to be accounted for. Their definitions and all missing proofs and examples are in the appendix.

## 3. Ordinal Preferences

In this section, we first introduce ordinal preferences. Then, we show that the axiom of *proportionality for solid coalitions* can be satisfied robustly in the incremental setting, but not in the decremental setting. We further give asymptotic lower bounds for the number of replacements that are required to restore proportionality after a single deletion, and present negative results for a notion called "rank justified representation".

An ordinal preference profile $P = (\succ_i)_{i \in N}$ is a collection of complete, strict orders $\succ_i$ over $C^*$. We hence write $\succ$ instead of $P$ in this section. We define $P|_C = \succ|_C$ as the restriction of each $\succ_i$ to $C$. Given a profile $\succ$ and a set $C \subseteq C^*$, we say that a set of voters $N' \subseteq N$ is a *solid coalition* over a set $S \subseteq C$ of candidates if for any $i \in N'$ it holds that $S \succ_i C \setminus S$, i.e., $S$ forms a prefix of that voter's preferences. A committee $W$ satisfies *Proportionality for Solid Coalitions (PSC)* (Dummett, 1984) if for every $\ell$-large group of voters $N'$ that is a solid coalition over some $S \subseteq C$, it holds that $|W \cap S| \geq \min(|S|, \ell)$. If $W$ violates PSC due to an $\ell$-large solid coalition $N'$ over $S$ with $|W \cap S| < \min(\ell, |S|)$, we say that $(N', S, \ell)$ (or $N'$) is a *witness* for the violation. PSC is a thoroughly studied axiom and, e.g., is satisfied by the popular and widely used single transferable vote (STV). The eminent social choice theorist Tideman argued for PSC as the reason for STV being proportional: "It is the fact that STV satisfies PSC that justifies describing STV as a system of proportional representation." (Tideman, 1995, page 28)

As an illustration of PSC consider the following example with $n = 4$ voters and $k = 3$:

$$c_1 \succ_1 c_2 \succ_1 c_3 \succ_1 c_4 \succ_1 c_5$$
$$c_3 \succ_2 c_1 \succ_2 c_2 \succ_2 c_5 \succ_2 c_4$$
$$c_4 \succ_3 c_1 \succ_3 c_5 \succ_3 c_3 \succ_3 c_2$$
$$c_3 \succ_4 c_2 \succ_4 c_1 \succ_4 c_4 \succ_4 c_5.$$

Here, the committee $W = \{c_2, c_3, c_4\}$ satisfies PSC. However, $W' = \{c_3, c_4, c_5\}$ violates PSC due to the witness $(\{1, 2, 4\}, \{c_1, c_2, c_3\}, 2)$, as $|W' \cap \{c_1, c_2, c_3\}| < 2$.

As our first main result, we give a polynomial time incremental rule satisfying PSC. Our proof entails an even stronger statement: for each committee $W$ satisfying PSC and each newly added candidate $c$ that causes a violation, there is always a single candidate $c' \in W$ who can be swapped with $c$ to restore PSC. Therefore, it is possible to, for instance, initialize the committee by running STV and then fix PSC violations whenever they appear.

**Theorem 3.1.** *There exists a robust incremental algorithm satisfying PSC.*

*Proof.* Fix an ordinal profile $\succ$ and target size $k$. Let $C_t = C_0 \cup \{c_1, \ldots, c_t\}$ denote the set of candidates that joined until time-step $t$. We will denote by $f(t)$ the committee selected at this time. For $t = 0$, let $f(t)$ be any committee satisfying PSC.

For $t > 0$, let $f(t-1)$ satisfy PSC with regard to $C_{t-1}$. We will show that $f(t-1)$ still satisfies PSC at time-step $t$, or that by adding $c_t$ to $f(t-1)$ and removing another candidate, PSC can be restored. To this end, we consider the set $W = f(t-1) \cup \{c_t\}$ and the resulting committees $W_{-c} = W \setminus \{c\}$ for $c \in W$. Rephrased, our goal is to show that $W_{-c}$ still satisfies PSC at time $t$ for some $c \in W$. Note that $c$ here can also be the newly added candidate $c_t$.

To reach this goal, we conduct a counting argument: for each $c \in W$ with $W_{-c}$ violating PSC, we "reserve" a unique $\frac{1}{k}$-fraction of the voters. As there are $k + 1$ candidates in $W$, there hence exists at least one candidate for which we could not have reserved such a $\frac{1}{k}$-fraction and thus no PSC violation occurs when we remove this candidate from $W$.

To show this, we need the two following claims about potential PSC witnesses. Let $c \in W$ be such that $W_{-c}$ violates PSC at time $t$ due to some witness $(N_c, S, \ell)$.

**Claim 1:** $c \in S$, i.e., all voters in $N_c$ rank $c$ above $C_t \setminus S$.

To prove the claim, we consider a case distinction:

*Case 1* is that $c_t \notin S$, which implies that the voter set $N_c$ is a solid coalition for the candidate set $S \subseteq C_{t-1}$ in time-step $t - 1$. Further, of course, $N_c$ is $\ell$-large independent of the time-step. Since $f(t-1)$ satisfies PSC at time $t - 1$, we have $|f(t-1) \cap S| \geq \min(|S|, \ell)$. By choice of the witness $(N_c, S, \ell)$, we have $|W_{-c} \cap S| < \min(|S|, \ell)$. This implies $|W_{-c} \cap S| < |f(t-1) \cap S|$, which is only possible if $c \in S$, showing our claim in this case.

*Case 2* is that $c_t \in S$, which implies that the voter set $N_c$ is a solid coalition for the candidate set $S \setminus \{c_t\} \subseteq C_{t-1}$ in time-step $t - 1$ (and still is $\ell$-large). Since $f(t-1)$ satisfies PSC at time $t - 1$, we have $|W \cap S| - 1 = |f(t-1) \cap (S \setminus \{c_t\})| \geq \min(|S| - 1, \ell)$, where the equal sign follows from $c_t \in W \cap S$ and $c_t \notin f(t-1)$. This implies $|W \cap S| \geq \min(|S| - 1, \ell) + 1 \geq \min(|S|, \ell) > |W_{-c} \cap S|$, which is only possible if $c \in S$.

**Claim 2:** $|W \cap S| \leq \ell$.

By choice of the witness $(N_c, S, \ell)$, we have $|W_{-c} \cap S| < \min(|S|, \ell) \leq \ell$. This directly implies $|W \cap S| \leq |W_{-c} \cap S| + 1 \leq \ell$, proving Claim 2.

Now we are ready for the main argument. Assume for contradiction that for all $c \in W$, the committee $W_{-c}$ does

not satisfy PSC. Fix any $c \in W$. Then, there exist witnesses $(N', S, \ell)$ of $W_{-c}$ violating PSC and we can exhaustively list all of them. Consider only the witnesses for $W_{-c}$ with maximal candidate set size $|S|$, and among these choose a witness that maximizes the size of the solid coalition $|N|$. For each $c \in W$, we denote this "maximal" witness via $(N_c, S_c, \ell_c)$. We will now arrive at a contradiction by iteratively assigning a $\frac{1}{k}$-fraction of the voters to each $c$ in $W$. For this, enumerate $W = \{d_1, \ldots, d_{k+1}\}$. For $d_1$, we can assign it to any $\frac{1}{k}$-fraction of the voters that is contained in $N_{d_1}$, which must exist since $N_{d_1}$ must be at least 1-large to be a witness. For $j \geq 2$, let injective voter assignments be made for all candidates with smaller index, i.e., $d_x$ with $1 \leq x < j$. To prove that there is still at least a $\frac{1}{k}$-fraction of the voters from $N_{d_j}$ unassigned, we use the following claim to count the number of $d_x$ with assignments in $N_{d_j}$.

**Claim 3:** For all $x < j$ with $N_{d_x} \cap N_{d_j} \neq \emptyset$, we have $N_{d_x} = N_{d_j}$ and $S_{d_x} = S_{d_j}$.

Let $i \in N_{d_x} \cap N_{d_j}$ be given. Since $S_{d_x}$ and $S_{d_j}$ are both prefixes of the relation $\succ_i$, one must be a subset of the other. Without loss of generality, assume $S_{d_x} \subseteq S_{d_j}$. We first show that the voter set $N_{d_j}$ is not only a witness for a PSC violation of $W_{-d_j}$, but also for a violation of $W_{-d_x}$. Applying Claim 1, we obtain $d_x \in S_{d_x} \subseteq S_{d_j}$ and $d_j \in S_{d_j}$. Combining this fact with $(N_{d_j}, S_{d_j}, \ell_{d_j})$ being a witness yields

$$\min(|S_{d_j}|, \ell_{d_j}) > |S_{d_j} \cap W_{-d_j}| = |S_{d_j} \cap W_{-d_x}|.$$

Hence, $(N_{d_j}, S_{d_j}, \ell_{d_j})$ is also a witness for the PSC violation of $W_{-d_x}$. Since we chose $S_{d_x}$ to be maximal, it must be that $S_{d_x} = S_{d_j}$ and the prefixes coincide. Thus, $(N_{d_j} \cup N_{d_x}, S_{d_x}, \ell_{d_x})$ is also a witness for the PSC violation of $W_{-d_x}$. Since we chose $N_{d_x}$ to be inclusion maximal among witnesses, it must be that $N_{d_x} = N_{d_j} \cup N_{d_x}$, implying $N_{d_x} \supseteq N_{d_j}$. Applying the same argument to $W_{-d_j}$, we obtain $N_{d_j} \supseteq N_{d_x}$ which concludes the proof of Claim 3.

Each $d_x$ with $x < j$ that is assigned some voters in $N_{d_j}$ satisfies $N_{d_x} \cap N_{d_j} \neq \emptyset$. By Claim 1 and 3, it follows that $d_x \in S_{d_j} \cap W$. However, by Claim 2, $|W \cap S_{d_j}| \leq \ell_{d_j}$. Since $d_j$ itself is contained in $W \cap S_{d_j}$, at most $\ell_{d_j} - 1$ different $d_x$ with $x < j$ can have been assigned to some voters in $N_{d_j}$. Since each of these $d_j$ is assigned to $\frac{n}{k}$ voters and $|N_{d_j}| \geq \ell_{d_j} \frac{n}{k}$, there is still at least a $\frac{1}{k}$-fraction of the voters unassigned, which we can assign to $d_j$. This proves that we can assign each $c \in W$ to some $\frac{1}{k}$-fraction of $N$. Since $W$ contains $k+1$ candidates, this implies that there are $\frac{k+1}{k} \cdot n > n$ voters in total, the desired contradiction. Hence, there exists a $c \in W$ such that $W_{-c}$ satisfies PSC. □

The approach of Theorem 3.1 does not work for deleting a candidate—if we choose poorly, we may need to replace the entire committee after a single deletion.

*Example* 1. A size $k$ committee satisfying PSC may require $k$ additions to restore PSC after a single deletion. For this, consider the following profile with $n = k$.

$$a_1 \succ_1 b_1 \succ_1 a_2 \succ_1 \cdots \succ_1 a_k \succ_1 *$$
$$a_1 \succ_2 b_2 \succ_2 a_2 \succ_2 \cdots \succ_2 a_k \succ_2 *$$
$$\vdots$$
$$a_1 \succ_k b_k \succ_k a_2 \succ_k \cdots \succ_k a_k \succ_k *$$

The committee $W = \{a_1, \ldots, a_k\}$ satisfies PSC. However, after removing $a_1$, all candidates $b_i$ would need to be added.

Similarly, commonly used rules that satisfy PSC fail to distinguish between robust and non-robust committees. We provide an example in the appendix.

*Observation* 1. The single transferable vote (STV) and the expanding approvals rule (EAR) of Aziz and Lee (2020) can select committees that are not robust to a single deletion for PSC, even when such committees exist.

While in Observation 1 a robust committee exists, there are instances where *any* algorithm satisfying PSC is not robust and requires at least $\Omega(\log(\log(k)))$ changes to restore PSC. This precludes the existence of a robust decremental—let alone fully-dynamic—PSC algorithm.

**Proposition 3.2.** *There does not exist a robust decremental PSC algorithm. After a single deletion, any algorithm may require $\Omega(\log(\log(k)))$ replacements to restore PSC.*

Finally, in the appendix, we also show that for the "rank-based" axioms recently introduced by Brill and Peters (2023) an even stronger lower bound of $\Omega(\sqrt{k})$ holds and that this lower bound also extends to incremental setting.

**Theorem 3.3.** *There is no incremental or decremental algorithm satisfying the rank-JR axiom of Brill and Peters (2023) and making $o(\sqrt{k})$ changes amortized per round.*

## 4. Proportional Clustering

Secondly, we turn to *proportional clustering*. Here, candidates and voters are situated in a space $(N \cup C^*, d)$ with $d$ being a *pseudometric*. Hence, the distance function $d \colon N \cup C^* \times N \cup C^* \to \mathbb{R}_{\geq 0}$ satisfies symmetry: $d(x, y) = d(y, x)$ and the triangle inequality: $d(x, y) \leq d(x, z) + d(z, x)$ for all voters and candidates $x, y, z \in N \cup C^*$. For finite $C \subseteq C^*$, we denote by $d|_C$ the natural restriction of $d$ to the finite subdomain $N \cup C \times N \cup C$. In proportional clustering, the most prominent proportionality notion is that of proportional fairness (Chen et al., 2019). An outcome (or clustering or committee) $W$ is said to be $\gamma$-*proportionally fair* for some $\gamma \geq 1$ and some instance $(C, d|_C, k)$, if there is no unselected candidate $c \in C \setminus W$ and 1-large group $N' \subseteq N$ of voters such that

$$\min_{c' \in W} d(i, c') > \gamma \, d(i, c) \quad \text{for all } i \in N'.$$

Proportional fairness, however, only looks at deviations to single candidates. This prompted several strengthenings (Ebadian and Micha, 2025; Kalayci et al., 2024; Aziz et al., 2024) dealing with deviations to multiple candidates instead. As it is closest in spirit to proportional fairness, we focus here on the *q-core* as introduced by Ebadian and Micha (2025). Given an instance over some candidate set $C \subseteq C^*$, a committee $W$ is said to be in the $\alpha$-$q$-core for some $q \in [k]$ and $\alpha \geq 1$ if for all other candidate subsets $C' \subseteq C$ the following holds: there are strictly less than $\frac{|C'|}{k} n$ voters $i \in N$ for which their $q$-th closest candidate in $W$ is $\alpha$-times farther away than their $q$-th closest candidate in $C'$. For $q = 1$ this is equivalent to proportional fairness.

Unlike for the ordinal and approval settings, in the clustering setting proportionality can only be approximated, not perfectly satisfied. Several existing algorithms, such as Greedy Capture (Chen et al., 2019) or the Spatial Expanding Approvals Rule (Aziz et al., 2024), achieve a constant factor approximation to proportional fairness or the core. However, these algorithms rely on a generalization of rank-JR — of which we have shown that it does not admit a robust algorithm — to proportional clustering (Kellerhals and Peters, 2024). We circumvent this and design a fully-dynamic algorithm achieving a constant factor approximation to proportional fairness. In essence, proportional clustering is easier than proportional multiwinner voting, as voters share a metric space with the candidates. By clustering similar voters into groups, we preempt the clustering of candidates. Using this, we obtain a fully-dynamic algorithm that is 4.24-proportionally fair and in the 5-$q$-core. Surprisingly enough, the 5-$q$-core bound is equal to the current best bound in the offline setting (Kellerhals and Peters, 2024).

**Theorem 4.1.** *There exists a robust fully-dynamic algorithm achieving a $2 + \sqrt{5} \sim 4.24$-proportional fair outcome and satisfying the 5-$q$-core for any $q \in [k]$.*

*Proof.* We begin with a pre-clustering phase similar to the spatial expanding approvals rule (Aziz et al., 2024). We assign each voter $i \in N$ a budget $b_i = \frac{k}{n}$ and initialize a counter $x = 1$. Then we continuously increase a real parameter $\delta$ from 0 on. Whenever there is a set of voters $N' \subseteq N$ of diameter at most $\delta$ such that $N'$ has a total budget of at least 1, we create a cluster $N_x = N'$ for these voters, decrease their budgets by a total of 1, and increase the counter $x$ by 1. If multiple clusters could be created for the same $\delta$ but compete with each other for the budget, we break ties arbitrarily. This leads to clusters $N_1, \ldots, N_k$ (with clusters potentially overlapping). To focus on the voters that positively contributed to the clusters, we denote by $p_i(x) \geq 0$ the contribution of voter $i$ to the cluster $N_x$.[1]

---

[1] This algorithm for "pre-clustering" the voters can additionally be seen as a generalization of the recently proposed algorithm for finding a proportional non-centroid clustering by Caragiannis et al.

For a given cluster of voters $N_x$ and any candidate $c$ we let $d(N_x, c) = \min\{d(i, c) \colon i \in N \text{ and } p_i(x) > 0\}$. To initialize a committee, we consider the clusters $N_1, \ldots, N_k$ in order. For each cluster $N_x$ we pick among the so far unchosen candidates some $c_x$ minimizing $d(N_x, c_x)$. To dynamically determine whether we need to modify the committee at each time step, we say that a cluster $N_x$ *envies* another cluster $N_y$ if $d(N_x, c_y) < d(N_x, c_x)$.[2] By our enumeration, it is clear that for the initial assignment and all $x, y \leq n$, cluster $N_y$ can only envy $N_x$ if $x < y$. Further, the selected committee contains the distance minimizers for all clusters $N_1, \ldots N_k$. We will in each step ensure that this invariant remains true.

When we delete a candidate $c$ we either do (i) nothing if $c$ is not picked by any cluster or (ii) if $c$ is picked by cluster $N_x$ we let $N_x$ repick the closest current unchosen candidate $d$. Since every cluster $N_y$ with $y \neq x$ prefers their current candidate to $d$, we re-enumerate them with $N_x$ is renamed to $N_k$ and the former $N_{x+1}, \ldots, N_k$ renamed to $N_x, \ldots, N_{k-1}$. Still, for $x, y \leq n$, $N_y$ can only envy $N_x$ if $x < y$.

If a candidate $c$ gets added, we first check if there is any cluster $N_x$ for which $d(N_x, c) < d(N_x, c_x)$, i.e., $c$ is closer than the picked candidate for this cluster. If there is not, we do not add $c$. If there is, however, such a cluster $N_x$, we assign $c$ to such a cluster $N_{x^*}$ with the smallest index $x^*$ and store $c_{x^*}$ as the *interim candidate*. As long as the set of $N_y$ with $y > i^*$ are assigned their current choices or $c_{x^*}$, $N_{x^*}$ only envies clusters with $y' < i$. Now, in ascending order, every $N_y$ with $y > i$ gets to choose between keeping their current candidate $c_y$ or exchanging it for the current interim candidate, thus making $c_y$ the new interim candidate. If $N_y$ keeps $c_y$, then it will not envy any $N_z$ with $z > y$ after the process since they will be assigned to some $c_{>y}$ or the interim candidate that $N_y$ prefers less to $c_y$. If $N_y$ swaps $c_y$ for the interim candidate, then it still will not envy any $N_{>y}$ after the process because each will obtain some $c_{\geq y}$. After $N_k$ made their choice, we discard the current interim candidate and obtain a new committee containing $c$ with just one swap on the currently selected set, but up to $k$ re-assignments of clusters to selected candidates. Still, this time without re-enumeration, for $x, y \leq n$, $N_y$ can only envy $N_x$ if $x < y$.

Now we are ready to show that this procedure is always $\rho$-proportional fair. Let $W$ be any committee throughout the online process, $c$ be any unselected candidate, and $N'$ be a group of voters deviating to $c$ of size at least $\frac{n}{k}$. Let $\delta$ be the diameter of $N'$. Then, in the first step of the procedure,

we know that at least one of the agents in $N'$ pays for a preprocessed cluster $N''$ of diameter at most $\delta$. Let $i$ be this agent and let $j$ be the agent furthest away from $c$ in $N'$ (note that this might be $i$ again). Since $N''$ did not pick $c$, $c$ is not a (unique) minimizer of $d(N'', \cdot)$. We know that there must be an agent $h \in N''$ with $d(h, W) \leq d(N'', c) \leq d(i, c)$. Further, by the triangle inequality we can bound the distance between any two elements of $N'$ by their respective distances to $c$, so it must hold that $\delta \leq 2d(j, c)$. Thus, we get that

$$
\begin{aligned}
\rho \leq \quad & \min\left(\frac{d(i, W)}{d(i, c)}, \frac{d(j, W)}{d(j, c)}\right) \\
\leq & \min\left(\frac{d(i, h) + d(h, W)}{d(i, c)}, \frac{d(j, i) + d(i, W)}{d(j, c)}\right) \\
\leq & \min\left(\frac{\delta + d(i, c)}{d(i, c)}, \frac{d(i, c) + d(j, c) + d(i, W)}{d(j, c)}\right) \\
\leq & \min\left(\frac{2d(j, c) + d(i, c)}{d(i, c)}, \frac{3d(j, c) + 2d(i, c)}{d(j, c)}\right) \\
\leq & \min_{x \geq 0}\left(2x + 1, 3 + \frac{2}{x}\right) = 2 + \sqrt{5}.
\end{aligned}
$$

and therefore the improvement through $c$ is bounded by $2 + \sqrt{5}$.

For the $\alpha$-$q$-core, let $N' \subseteq N$ be an $\ell$-large deviating coalition and $C' \subseteq C$ of size $|C'| = \ell$ be the set of candidates the coalition deviates to. For a given agent $i \in N$ we define $d^q(i, C')$ to be the distance of the agent to their $q$-th closest member of $C'$. Following Kellerhals and Peters (2024, Lemma 9) there is a candidate $c \in C'$ and a subset $N'' \subseteq N$ such that $c$ is in the top-$q$ choices among $C'$ of everyone in $N''$ with $N''$ being of size at least $q\frac{n}{k}$. Let $i \in N''$ be the agent among $N''$ with the largest $d^q(i, C')$. Since the agents in $N''$ have a total budget of $q$ there must at least exist $q$ clusters bought partially by agents from $N''$. Similar to the first part of this proof, the diameter of at least $q$ of these clusters is smaller than the diameter of $N''$. We know that these clusters must have chosen $q$ of the cluster centers in $W$. Since the diameter of $N''$ is at most $d(i_1, i_2) \leq d(i_1, c) + d(c, i_2) \leq d^q(i_1, C') + d^q(i_2, C') \leq 2d^q(i, C')$ the diameter of each of these clusters is also at most $2d^q(i, C')$. Since each agent in $N''$ is also at most $d^q(i, C')$ away from an unselected cluster center, the cluster center selected must also be at most $d^q(i, C)$ away from someone in the group and thus at most $3d^q(i, C')$ away from the member of $N''$. However, since the diameter of $N''$ is at most $2d^q(i, C')$ this implies that agent $i$ is at most a distance of $5d^q(i, C')$ away from $q$ cluster centers in $W$, therefore showing that $W$ is in the $5$-$q$-core. $\square$

While the initial pre-clustering is not tractable, it is possible to adapt this step to obtain a polynomial time constant factor approximation with slightly larger constant.

---

(2024) to also handle groups of larger sizes than $\frac{n}{k}$.

[2]We now perform a procedure similar to the envy-cycle elimination from fair division, see, e.g., the survey of Amanatidis et al. (2023).

# 5. Approval Preferences

In this section, we assume that we are given an *approval profile* $A = (A_i)_{i \in N}$, and we call $A_i \subseteq C^*$ the *approval ballots*. We define $A|_C = (A_i \cap C)_{i \in N}$. Throughout this section, we refer to subsets $W \subseteq C$ of size $|W| \leq k$ as *subcommittees*. For approval preferences, we call a set $N' \subseteq N$ of voters $\ell$-cohesive if $|\bigcap_{i \in N'} A_i| \geq \ell$. Since the original paper of Aziz et al. (2017) defining the axiom of *justified representation*, a plethora of works have focused on defining new possible axioms or extending the original ones of Aziz et al. (see for instance the recent book of Lackner and Skowron, 2023). We focus here on two natural variants introduced by Brill and Peters (2023). We say that a committee $W$ satisfies

- *proportional justified representation+ (PJR+)* if for every $\ell \in [k]$ and 1-cohesive and $\ell$-large group $N'$ it holds that $\bigcap_{i \in N'} A_i \subseteq W$ or $|\bigcup_{i \in N'} A_i \cap W| \geq \ell$.

- *extended justified representation+ (EJR+)* if for every $\ell \in [k]$ and 1-cohesive and $\ell$-large group $N'$ it holds that $\bigcap_{i \in N'} A_i \subseteq W$ or $|A_i \cap W| \geq \ell$ for some $i \in N'$.

Further, $\alpha$-EJR+ with $\alpha > 1$ requires that for every $\ell \in [k]$ and 1-cohesive and $\alpha \cdot \ell$-large group $N'$ there exists an $i \in N'$ with $|A_i \cap W| \geq \ell$ or it holds that $\bigcap_{i \in N'} A_i \subseteq W$. *Example* 2. First, we give an example of a committee satisfying EJR+, for which removing any candidate from the committee, forces at least two changes, even for PJR+ to be satisfied. For this, consider the following simple instance with $k = 2$, two voters, and respective approval sets $\{a, b\}$ and $\{b, c\}$. Further, there is a candidate $d$ approved by no one. The committee $\{b, d\}$ satisfies EJR+, but after the removal of $b$, only the committee $\{a, c\}$ satisfies PJR+.

A similar incremental example can also be created. Consider an instance with $k = 3$ and six voters having approval sets $\{a, b\}, \{a, c\}, \{a, d\}, \{a, e\}, \{f\}, \{f\}$. Consider the committee $\{b, c, d\}$ before $f$ becomes feasible. After the addition of $f$ to the instance, we would need to remove one of $b, c, d$, for whom the voter approving it, together with the voter approving $\{a, e\}$ would witness a PJR+ violation.

**PJR+** Nonetheless, we now construct robust incremental and almost robust fully dynamic PJR+ algorithms. We require the following notion by Brill and Peters (2024), which is related to *priceability* (Peters and Skowron, 2020).

A subcommittee $W \subseteq C$ of size at most $k$ is *maximally affordable* if there exists a payment system $(p_i)_{i \in N} : C \to \mathbb{R}_{\geq 0}$ satisfying the following constraints

**C1** $p_i(c) = 0$ if $c \notin A_i$ for all $c \in C$ and $i \in N$

**C2** $\sum_{c \in C} p_i(c) \leq \frac{k}{n}$ for all $i \in N$

**C3** $\sum_{i \in N} p_i(c) = 1$ for all $c \in W$

**C4** $\sum_{i \in N} p_i(c) = 0$ for all $c \notin W$

**C5** $\sum_{i \in N : c \in A_i} \left( \frac{k}{n} - \sum_{d \in C} p_i(d) \right) < 1$ for all $c \notin W$.

Maximally affordable subcommittees always exist, as e.g., the non-exhaustive output of the MES rule is maximally affordable. Further, it is easy to see that any committee $W'$ that contains a maximally affordable subcommittee $W \subseteq W'$ satisfies PJR+ (Brill and Peters, 2023, Proposition 10).

**Corollary 5.1.** *Every completion of a maximally affordable subcommittee satisfies PJR+.*

Our goal is to maintain supersets of maximally affordable subcommittees dynamically. First, we use this approach to provide a robust incremental PJR+ algorithm, which, in essence, works the same way as the algorithm of Do et al. (2022), who show that PJR+ is satisfiable in their online committee selection setting.

**Theorem 5.2.** *There exists a robust incremental PJR+ algorithm.*

The proof also shows that once a maximally affordable committee of size $k$ has been instantiated, no further modifications are required to maintain PJR+ in all time steps.

Using affordability, however, does not yield a robust decremental PJR+ algorithm. In fact, no such algorithm exists.

**Theorem 5.3.** *There does not exist a robust decremental PJR+ algorithm.*

Nonetheless, we can utilize maximally affordable committees to construct a fully-dynamic PJR+ algorithm making two changes per iteration *amortized*, meaning that the total number of changes up to any time-step $t$ is at most $2t$.

**Theorem 5.4.** *There exists a robust fully-dynamic PJR+ algorithm making amortized 2 changes per iteration.*

**EJR+** While our previous result shows that one can nearly achieve PJR+ in a fully-dynamic manner, PJR+ in itself is quite a weak axiom for approval-based multiwinner voting (see for instance Peters and Skowron, 2020, Example 6). The strongest alternative to PJR+ which is still achievable in polynomial time is EJR+. However, EJR+ is not nearly as well understood as PJR+, which is significantly easier to achieve. Consequentially, fewer rules are known to satisfy EJR+. As any committee satisfying EJR+ also satisfies PJR+, our impossibility result in Theorem 5.3 also applies to EJR+.

As our first positive result we build upon an approximation result from Do et al. (2022) and show that a $\mathcal{O}(\log(k))$-approximation of EJR+ is possible in a fully-dynamic setting

making amortized two change per iteration. This in essence works similar to Theorem 5.4.

**Theorem 5.5.** *There exists a fully-dynamic* $\Theta(\log(k))$*-EJR+ algorithm making amortized two changes per iteration.*

While Theorem 5.5 considers fully dynamic algorithms, it only provides an approximation factor depending on $k$. If we turn to the incremental setting, we can obtain $\alpha$-EJR+ bounds for constant $\alpha > 1$. Especially, the following result implies an incremental algorithm satisfying 2-EJR+ making amortized 2 changes per iteration.

**Theorem 5.6.** *For any* $\alpha > 1$ *there exists an incremental* $\alpha$*-EJR+ algorithm making amortized* $\frac{\alpha}{\alpha-1}$ *changes.*

While achieving a robust algorithm satisfying exact EJR+ seems difficult, we are able to provide a committee that is robust with respect to a single added candidate. That is, for any committee $W$ selected by the rule, and any added candidate $c'$ there exists a candidate $c \in W \cup \{c'\}$ such that $(W \cup \{c'\}) \setminus \{c\}$ satisfies EJR+. This rules out strong negative results for incremental algorithms, akin to Proposition 3.2 for PSC in the decremental setting. We achieve this by modifying the *greedy justified candidate rule* (GJCR) of Brill and Peters (2023) to be "locally stable".

**Theorem 5.7.** *There exists an incremental EJR+ algorithm that is robust with respect to a single addition.*

The modification is necessary even for this single step, as we give examples in the appendix that all known rules satisfying EJR+ are not robust with respect to a single addition. This leads us to the open question whether there is an incremental algorithm for EJR+ that remains robust beyond the first step.

**Open Question 1.** *Is there a robust (amortized) incremental algorithm satisfying EJR+?*

We further remark that all commonly considered rules satisfying EJR+ fail to distinguish between decrementally robust and unrobust committees.

*Example* 3. Consider the profile with $2 \times \{a_1, \ldots, a_5, x\}$, $2 \times \{a_1, \ldots, a_5, y\}$, $1 \times \{a_1, \ldots, a_5\}$, $2 \times \{b_1, \ldots, b_4, x\}$, $2 \times \{b_1, \ldots, b_4, y\}$, $1 \times \{c_1, \ldots, c_4, x\}$, $1 \times \{c_1, \ldots, c_4, y\}$, $2 \times \{c_1, \ldots, c_4\}$ and fix $k = n = 13$. Then, all rules known to satisfy EJR+ (MES, GJCR, and PAV) can choose the committee $\{a_1, \ldots, a_5\} \cup \{b_1, \ldots, b_4\} \cup \{c_1, \ldots, c_4\}$. However, after the deletion of some $a_i$, we would need to add both $x$ and $y$ to the committee to restore EJR+.

It remains unknown whether we can select an EJR+ committee that is robust to a single deletion, and whether there is a fully-dynamic or decremental algorithm satisfying a constant approximation to EJR+.

**Open Question 2.** *Is there an algorithm satisfying EJR+ that is robust with respect to a single deletion? Does there exist a fully-dynamic algorithm satisfying* $\mathcal{O}(1)$*-EJR+ making amortized* $\mathcal{O}(1)$ *changes per round?*

## 6. Conclusion and Open Questions

We study proportionality in multiwinner voting for dynamic candidate sets, with voter preferences given via linear orders, distances in a metric space, or approval ballots. Depending on the setting and whether new candidates join, old candidates drop out, or both, we provide algorithms that make small amounts of changes while upholding proportionality or show that no such algorithm can exists by providing a lower bound on the changes necessary. An overview of our results can be seen in Table 1.

Our work leaves open several questions and possible future research directions. As a "meta" future research direction, we highlight that the understanding of EJR and EJR+ as axioms is still quite narrow. For instance, despite extensive research, it is still an open question whether there always exists a ranking satisfying EJR for every prefix of the ranking (Skowron et al., 2017; Chandak et al., 2024) or whether there is a safe querying procedure for EJR (Brill et al., 2023). Furthering the understanding of EJR and EJR+, e.g., by developing different rules or characterizations of rules satisfying EJR+ might shed further light on it and help to resolve the open questions regarding EJR and EJR+.

As a further point, we restrict ourselves to dynamic *candidate* sets. A natural extension would be a setting in which not only the candidates, but also the voters are dynamic. This gives some additional difficulties. For instance, adding or deleting voters, changes the quota $\frac{n}{k}$ throughout the process. Deciding whether it is feasible to deal with this, is an interesting possibility for future work.

### Acknowledgements

Chris Dong was supported by the Deutsche Forschungsgemeinschaft under grants BR 2312/11-2 and BR 2312/12-1. Jannik Peters was supported by the Singapore Ministry of Education under grant number MOE-T2EP20221-0001.

### Impact Statement

Our paper is a theoretical work dealing with online algorithms ensuring "proportionality" and tries to enhance the understanding of proportional outcomes and the algorithms achieving them. While the goal of proportionality is to ensure that large groups of the affected population are adequately represented, naturally, this does not rule out that there are other potential outcomes that are more beneficial to society or "fairer" in some other aspects. In particular, it is very much unclear whether the algorithms we designed here should or could ethically be used for real-world elections.

| Axiom | Incremental | Decremental and Fully-Dynamic |
|---|---|---|
| PSC | ✓ | ✗ $\Omega(\log(\log(k)))$ lower b. |
| rank-JR | ✗ $\Omega(\sqrt{k})$ lower b. | ✗ $\Omega(\sqrt{k})$ lower b. |
| Proportional Fairness | ✓4.24-approx | ✓4.24-approx |
| $q$-Core | ✓5-approx | ✓5-approx |
| PJR+ | ✓ | ✗,  ✓(amort. 2-changes) |
| EJR+ | ✓2-approx (amort. 2-changes) | ✓$\mathcal{O}(\log(k))$-approx (amort. 2-changes) |

*Table 1.* An overview of our results. A ✓ indicates that a robust (amortized/approximately optimal) dynamic algorithm exists for the respective problem and input type. An ✗ indicates that no fully-dynamic algorithm can exist.

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

**Algorithm 1** Greedy Justified Candidate Rule (GJCR) (Brill and Peters, 2023)

---

$W \leftarrow \emptyset$ **for** $\ell$ *in* $k, \dots, 1$ **do**

    **while** *there is* $c \notin W$: $|\{i \in N_c : |A_i \cap W| < \ell\}| \geq \frac{\ell n}{k}$ **do**

        Add candidate $c$ maximizing $|\{i \in N_c : |A_i \cap W| < \ell\}|$ to $W$

    **end**

**end**

return $W$

---

# A. Appendix: Multiwinner Voting Rules

In this section, we provide brief definitions of the mentioned rules.

**Rules for approval ballots.** *Proportional approval voting* (PAV) is the rule that assigns a score of $H(|A_i \cap W|)$ to each ballot $A_i$ and committee $W$, where $H(j)$ is the $j$-th harmonic number. It then chooses the committees that maximize the sum of scores over the entire profile $A$.

The *method of equal shares* (MES) assigns each voter a budget of $\frac{k}{n}$. Each candidate can be bought into the committee for a cost of $1$. MES proceeds iteratively and selects the next candidate by maximizing the minimal budget over the buyers after the purchase.

The *greedy justified candidate rule* (GJCR) also proceeds iteratively, by choosing the candidate currently causing the largest violation of EJR+. See Algorithm 1.

**Rules for ranked ballots.** *Single transferable vote* (STV) selects the candidates iteratively. Fixing $n$ as the initial number of voters and starting with an empty committee, it adds any candidate $c$ that is top ranked at least $\frac{n}{k}$ times. (It does not matter which one if there are multiple such candidates.) STV then deletes this share of supporters of $c$ and $c$ itself from the profile. In case that no candidate is top ranked sufficiently many times, instead a candidate with the lowest amount of supporters is eliminated from the profile.

The *Expanding approvals rule* (EAR) also assembles the committee step by step. It assigns each voter a budget of $1$ and proceeds rank by rank. If there is a candidate, for whom the voters giving it at most that rank have a budget of at least $\frac{n}{k}$, that candidate is added to the committee and the budget of these voters is decreased by $\frac{n}{k}$. Otherwise, the rank gets increased by $1$.

# B. Missing Proofs for Section 3

We examine the notion of rank-JR and present the missing proofs for PSC.

## B.1. Rank Justified Representation

A voter $i$ has rank $\mathrm{rank}(i, c)$ for candidate $c$ if this candidate is the $\mathrm{rank}(i, c)$th closest candidate to this voter. An outcome $W$ satisfies *rank justified representation (rank-JR)*, if there is no rank $r \leq m$, no 1-large set of voters $N' \subseteq N$ and set of candidates $C' \subseteq C$ such that $\mathrm{rank}(i, c) \leq r$ for all $i \in N'$ and $c \in C'$ as well as $|\{c \in W : \mathrm{rank}(i, c) \leq r$ for some $i \in N'\}| \cap W = \emptyset$. In the following example with $n = 4$ voters and $k = 2$, all committees satisfy PSC, but due to $r = 2$, only committees containing $b$ satisfy rank-JR (Brill and Peters, 2023).

$$a \succ b \succ e \succ d \succ c$$
$$d \succ b \succ c \succ e \succ a$$
$$c \succ b \succ e \succ d \succ a$$
$$e \succ b \succ d \succ c \succ a$$

For rank-JR, we show that neither incremental nor decremental rules can be robust.

**Theorem B.1.** *There is no robust incremental or decremental rank-JR algorithm.*

*Proof.* We give the following example with $n = 12$ voters, $m = 13$ candidates $C = \{c_1, \dots, c_6, c'_1, \dots, c'_6, \bar{c}\}$, and $k = 6$. We show that the removal of $c_1$ from $C$ is incompatible with decremental rank-JR and the addition of $c_1, c'_1$ to $C \setminus \{c_1, c'_1\}$ is incompatible with incremental rank-JR:

| | |
|---|---|
| 1. $c_1 \succ c_2 \succ \dots$ | 1'. $c'_1 \succ c'_2 \succ \dots$ |
| 2. $c_1 \succ c_3 \succ \dots$ | 2'. $c'_1 \succ c'_3 \succ \dots$ |
| 3. $c_2 \succ c_5 \succ \bar{c} \dots$ | 3'. $c'_2 \succ c'_5 \succ \bar{c} \dots$ |
| 4. $c_3 \succ c_4 \succ \bar{c} \dots$ | 4'. $c'_3 \succ c'_4 \succ \bar{c} \dots$ |
| 5. $c_5 \succ c_6 \succ \bar{c} \dots$ | 5'. $c'_5 \succ c'_6 \succ \bar{c} \dots$ |
| 6. $c_4 \succ c_6 \succ \bar{c} \dots$ | 6'. $c'_4 \succ c'_6 \succ \bar{c} \dots$ |

Note that this instance is symmetric with regard to ranking candidates $c_i$ and $c'_i$. We first show the following claim, which is crucial for both proofs:

**Claim 1:** If a committee $W$ satisfies rank-JR on $C$, then $|W \cap \{c_2, c_3, c'_2, c'_3\}| \leq 1$. To show this claim, first let a committee $W$ be given with $\{c_2, c_3\} \subseteq W$. Since $c_1, c'_1$ are top ranked twice, $c_1, c'_1 \in W$. Further, voters 5, 6 share candidate $c_6$ at position $r = 2$ and thus one of $\{c_4, c_5, c_6\}$ must be contained in $W$. Since $|W| = 6$, $W$ can only contain one of the candidates $\{c'_2, \dots, c'_6\}$. Thus, there are always $\{i', j'\} \subset \{3', \dots, 6'\}$ obtaining none of their three

most preferred candidates, leading to the desired violation of rank-JR for $W$ with $r = 3$. Now, let a committee $W$ be given with $|W \cap \{c_2, c_3\}| = 1$ and $|W \cap \{c'_2, c'_3\}| = 1$. We have to show that $W$ violates rank-JR. Again, $c_1, c'_1 \in W$. Further, voters $5, 6$ share candidate $c_6$ at position $r = 2$ and thus one of $\{c_4, c_5, c_6\}$ must be contained in $W$. Since the same argument holds for voters $5', 6'$, the committee $W$ already contains 6 candidates and $\bar{c} \notin W$. Note however, no matter how we choose the candidates, there will be one voter $i \in \{3, 4, 5, 6\}$ not obtaining any of their top three ranked candidates: If we choose $c_5$ $(c_4)$, voter 6 (voter 5) is unhappy. If we choose $c_6$, voter 3 or voter 4 will be unhappy as we can only choose one of $\{c_{2,3}\}$. By applying the same argument, we obtain $j \in \{3', 4', 5', 6'\}$ for which $W$ does not contain any of their top three candidates. Since $i, j$ both have $\bar{c}$ as their third most-preferred candidate, $\{i, j\}$ induces a rank-JR violation for $W$ and $r = 3$. This concludes the proof of the claim.

The following is clearly true: **Claim:** If a committee $W^2$ on $C \setminus \{c_1, c'_1\}$ satisfies rank-JR, then $\{c_2, c_3, c'_2, c'_3\} \subset W^2$.

For the incremental rank-JR violation, we see the following: Any rule $f$ satisfying rank-JR on $C^2 = C \setminus \{c_1, c'_1\}$ must return $W^2$ containing $\{c_2, c_3, c'_2, c'_3\}$. For $C^1 = C \setminus \{c'_1\}$, thus $|W^1 \cap \{\{c_2, c_3, c'_2, c'_3\}\}| \geq 3$ and thus for $C^0 = C$ we have $|W^0 \cap \{\{c_2, c_3, c'_2, c'_3\}\}| \geq 2$. This implies that $f$ fails rank-JR by our Claim 1.

For decremental rank-JR, apply Claim 1 to obtain that wlog $W \cap \{c_2, c_3\} = \emptyset$. Deleting $c_1$ now requires the addition of both $c_2, c_3$, as desired. $\qquad \square$

We generalize this approach to show that for any $t$ there is an instance in which at least $t$ changes must be made after deleting a single candidate. This provides steeper lower bounds than for PSC.

**Theorem 3.3.** *There is no incremental or decremental algorithm satisfying the rank-JR axiom of Brill and Peters (2023) and making $o(\sqrt{k})$ changes amortized per round.*

*Proof.* We now create a profile in which each committee satisfying rank-JR must add $t$ candidates $b^1, \dots b^t$ after the deletion of some candidate $a$ to maintain rank-JR. Consider the following instance consisting of $t$ blocks $(B_i)_{i \leq t}$, each $B_i$ consisting of $t(t + 1)$ voters with the following $3t$ voter types $(v_i^j)_{j \leq 3t}$:

$$1 \times v_i^1 : \quad a_i \succ b_i^1 \succ \dots$$
$$\dots$$
$$1 \times v_i^t : \quad a_i \succ b_i^t \succ \dots$$
$$(t-1) \times v_i^{t+1} : \quad b_i^1 \succ c_i^1 \succ \dots$$
$$\dots$$

$$(t-1) \times v_i^{2t} : \quad b_i^t \succ c_i^t \succ \dots$$
$$1 \times v_i^{2t+1} : \quad c_i^1 \succ d_i^1 \succ \bar{c} \succ \dots$$
$$\dots$$
$$1 \times v_i^{3t} : \quad c_i^t \succ d_i^t \succ \bar{c} \succ \dots$$

We set $k = t(t + 1)$ and thus $\frac{n}{k} = t$. For now, consider a fixed block $B_i$, $i \in [t]$. First, we notice that $a_i$ needs to be included to satisfy rank-JR, as there are $t$ voters top ranking it. To not need $t$ additions after the deletion of $a_i$, we include some $b_i^{x(i)}$ with $x(i) \leq t$ for each $i$. Further, we claim that $\bar{c}$ cannot be chosen: for each $i \in [t], j \in [t] \setminus \{x(i)\}$ to satisfy the $(t - 1)$ voters $v_i^{t+j}$ and the voter $v_i^{2t+j}$ we need to include one of $b_i^j, c_i^j, d_i^j$ to satisfy rank-JR. All additions so far considered for $B_i$, this enforces $t + 1$ candidates. Iterating over all blocks, we overall enforce $t(t + 1)$ candidates, filling the committee and leaving $\bar{c}$ unchosen. But then the set of $t$ voters $(v_i^{2t+x(i)})_{i \leq t}$ witness a rank-JR violation, as they all rank candidate $\bar{c}$ on rank 3.

For the incremental case consider the same profile before the addition of $(a_i)_{i \leq t}$

$$1 \times v_i^1 : \quad b_i^1 \succ \dots$$
$$\dots$$
$$1 \times v_i^t : \quad b_i^t \succ \dots$$
$$(t-1) \times v_i^{t+1} : \quad b_i^1 \succ c_i^1 \succ \dots$$
$$\dots$$
$$(t-1) \times v_i^{2t} : \quad b_i^t \succ c_i^t \succ \dots$$
$$1 \times v_i^{2t+1} : \quad c_i^1 \succ d_i^1 \succ \bar{c} \succ \dots$$
$$\dots$$
$$1 \times v_i^{3t} : \quad c_i^t \succ d_i^t \succ \bar{c} \succ \dots$$

Clearly, we must choose all $b_j^i$ for $i, j \leq t$, i.e., $t^2$ candidates. However, after adding $a_1, \dots, a_t$, we can similarly to the decremental case prove that at most $< t$ candidates of the form $b_j^i$ can be contained in a committee satisfying rank-JR. This constitutes $> t^2 - t$ replacements in $t$ rounds, hence in some round there must have been $t - 1 = \Omega(\sqrt{k})$ replacements. $\qquad \square$

This further raises the question whether the bound is tight.

**Open Question 3.** *Is there an incremental or decremental algorithm satisfying rank-JR making at most $\mathcal{O}(\sqrt{k})$ changes amortized per round?*

## B.2. PSC

*Observation* 1. The single transferable vote (STV) and the expanding approvals rule (EAR) of Aziz and Lee (2020) can

select committees that are not robust to a single deletion for PSC, even when such committees exist.

*Proof.* For an example of STV failing this, consider the following instance

$$a \succ b \succ \cdots$$
$$a \succ c \succ \cdots$$
$$b \succ a \succ x$$
$$x \succ y \succ \cdots$$
$$c \succ a \succ x$$
$$y \succ x \succ \cdots$$

Here, with $k = 3$ STV could first select $a$ and delete the first two voters who top-rank $a$ from the profile. Then delete $b$, select $x$, and delete voters three and four who at this time rank $x$ on top. Finally, delete $e$ and choose $y$. Now if $a$ withdraws, both $b$ and $c$ need to be added. We note that the same committee could also be selected by EAR. This instance, however, admits a robust committee, for instance $\{a, b, x\}$ is robust to a single deletion. $\square$

**Proposition 3.2.** *There does not exist a robust decremental PSC algorithm. After a single deletion, any algorithm may require $\Omega(\log(\log(k)))$ replacements to restore PSC.*

*Proof.* Let $t < s \in \mathbb{N}$ with $r = \binom{s}{t}$ Consider the following instance over the candidate set $C = \{a_1, \ldots a_r\} \cup \{b_1, \ldots, b_s\}$:

$$a_1 \succ b_1$$
$$\ldots$$
$$a_1 \succ b_t$$
$$a_2 \succ b_2$$
$$\ldots$$
$$a_2 \succ b_{t+1}$$
$$\ldots$$
$$a_r \succ b_{s-t+1} \succ \cdots$$
$$\ldots$$
$$a_r \succ b_s \succ \cdots$$
$$(t-1) \times b_1$$
$$\ldots$$
$$(t-1) \times b_s$$

To be more precise, for each subset of $t$ candidates from $B$, i.e., each $B' \in \binom{B}{t}$, we add one voter ranking $a_{i(B')}$ first and $b \in B'$ second. This yields $tr$ voters. Then, for each $i \leq s$ we add $t - 1$ voters ranking $b_i$ first. This adds

further $(t - 1)s$ voters. Whenever $s$ is a multiple of $t$, we can set $k = r + \frac{t-1}{t}s$. Thus, a set of voters is 1-large if it has cardinality $\geq t$. PSC now implies that $a_1, \ldots, a_r$ must all be chosen. Further, if in the next time step $a_i$ is deleted, then PSC enforces that all candidates $b$ in the subset $B'(i) \subset B$ are chosen. Since any $a_i$ could be deleted, this means that for the committee to guarantee PSC after one deletion, $s - 1$ candidates from $B$ must already be chosen in this round. Clearly, $s - 1 + r > \frac{t-1}{t}s + r = k$. Choose $s = t^2$. Then, $s - 1 + r - k = \frac{1}{t}s - 1 = t - 1$. So at least $t - 1$ candidates $b_i$ remain unchosen. If the corresponding $a_j$ gets deleted, at least $t - 2$ additions are necessary to restore PSC. For the asymptotic bound, observe $k \in \Theta(r) = \Theta(s^t)$. Thus, there is a constant $c \in \mathbb{R}_{>0}$ with $k \leq ct^{2t}$, which implies $\sqrt{\frac{k}{c}} \leq t^t$ and thus $\log(\sqrt{\frac{k}{c}}) \leq t \log(t)$, and finally $W(\log(\sqrt{\frac{k}{c}})) \leq t$ for the product log function $W$. Thus, the number of additions necessary to restore PSC have a lower bound of $\Omega(\log(\log(k)))$. $\square$

# C. Missing Proofs for Section 5

We first remark that the modification to GJCR is truly necessary to obtain a committee that is robust with respect to a single addition.

**Proposition C.1.** *GJCR, MES, and PAV can elect committees that are not robust with respect to a single addition.*

*Proof.* Consider the following approval profile with approval sets $1 \times \{a_1, \ldots, a_4\}, 3 \times \{a_1, \ldots, a_4, x\}, 2 \times \{b_1, \ldots, b_5, x\}, 3 \times \{b_1, \ldots, b_5\}, 1 \times \{c_1, \ldots, c_4\}, 3 \times \{c_1, \ldots, c_4, y\}, 2 \times \{d_1, \ldots, d_5, y\}, 3 \times \{d_1, \ldots, d_5\}$ with $n = 18 = k$. Here a possible MES committee is all candidates except for $x$ and $y$. However, now adding a candidate approved by the three voters voting for only $b$ candidates and the three voters only voting for $d$ candidates, would require one other candidate to be removed. If an $a$ or $c$ candidate is removed that candidate witnesses an EJR+ violation. If a $b$ or $d$ candidate gets removed, the corresponding voters approving $x$ or $y$ witness an EJR+ violation, as consist of 5 voters, but only approve 4 candidates in the outcome.

To extend this to PAV consider the same instance, consisting of 6 copies of $3 \times \{a_1, \ldots, a_4\}, 1 \times \{a_1, \ldots, a_4, x\}, 4 \times \{b_1, \ldots, b_5, x\}, 1 \times \{b_1, \ldots, b_5\}$ with $n = 54 = k$ Here, a optimal PAV committee would choose the copies of the $a$ and $b$ candidates, as adding the $x$ candidate for an $a$ candidate would decrease the PAV score by $-\frac{3}{4} + \frac{4}{6} < 0$ while adding the $x$ candidate for a $b$ candidate would leave the PAV score unchanged. However, adding a candidate approved exactly by the $\{b_1, \ldots, b_5\}$ voters, would need this candidate to be included, leading to the same contradiction as in the first case. $\square$

**Corollary 5.1.** *Every completion of a maximally affordable*

*subcommittee satisfies PJR+.*

*Proof.* Let $W = \{c_1, \ldots, c_\ell\}$ be maximally affordable w.r.t some $(p_i)_i$. Then clearly, $\ell \leq k$ as the voters only have budget $k$ in total. Assume for contradiction there is a violation of $PJR+$ for $W$ based on size-$k$ largeness, i.e., some $c \in C \setminus W$ with some $N' \subseteq N$ such that $c \in \bigcap_{i \in N'} A_i$ and $|\bigcup_{i \in N'} A_i \cap W| < \ell$ despite $|N'| \geq \ell \frac{n}{k}$ for some $\ell \in \mathbb{N}$. The total amount of budget spent by $N'$ is thus at most $\ell - 1$ with their starting budget being at least $\ell \frac{n}{k} \frac{k}{n} = \ell$. This is the desired contradiction as **C5<** is violated. $\square$

**Theorem 5.2.** *There exists a robust incremental PJR+ algorithm.*

*Proof.* The first $k$ candidates that arrive we take into our committee $W^0 = C^0$. We create a partition of $W^0 = X^0 \cup Y^0$ into a maximally affordable committee $X^0$ and a disposable part $Y^0$. For this, initialize both sets as empty and assign a total budget of $k$ equally among all voters, i.e., $\frac{k}{n}$ to every voter. They will proceed to buy candidates, each for the price of 1, into $X^0$ as follows: As long as there is a candidate $c$ such that $N_c$ has a total budget of $\geq 1$ to afford it, add $c$ to $X^0$, and subtract the budget of 1 off any of these voters in any way such that the budgets are not exceeded. Store this subtracted amount as $p_i(c)$. By finiteness of the budget, this process must terminate after a finite number of steps. Clearly, **C1** to **C4** are satisfied for $X^0$. Further, **C5<** is satisfied as else the candidate violating the inequality would be bought into the committee and the process cannot have already terminated. Clearly, $|X^0| \leq k$ as the voters only have budget $k$ in total. Now, set $Y^0 = C^0 \setminus X^0$. This concludes the induction start.

For the induction step, let two disjoint sets $X^t, Y^t \subseteq C^t$ given such that their union is of size $k$ and $X^t$ is maximally affordable in $C^t$ with cost function $p^t$ (and thus satisfies $PJR+$ with respect to size $k$ largeness). Let now a new candidate $c^*$ be added, i.e., $C^{t+1} = C^t \cup \{c^*\}$. If $X^t$ is still maximally affordable, we can set $X^{t+1} = X^t$, $Y^{t+1} = Y^t$, and $p^{t+1} = p^t$. Else, there must be $c \in C^{t+1}$ such that one of the conditions is violated. Since $C1$ to $C4$ did not change, it must be **C5<** and $c = c^*$. Thus, $N_{c^*}$ can afford to buy $c^*$ into the committee. Set $X^{t+1} = X^t \cup \{c^*\}$. Clearly, since there was budget left to buy $c^*$, we have $|X^t| < k$ and thus $|Y^t| > 0$. Thus, remove an arbitrary element $y \in Y^t$, i.e., set $Y^{t+1} = Y^t \setminus \{y\}$, Set $p_i(c^*)$ as the amount of budget that was taken from voter $i$ to finance $c^*$. It is easy to check that $W^{t+1}$ satisfies all 5 axioms and thus is maximally affordable. This concludes the induction step. $\square$

**Theorem 5.3.** *There does not exist a robust decremental PJR+ algorithm.*

*Proof.* Consider an instance with a set of 12 voters $N = \{1, \ldots, 12\}$, target size $k = 6$, and a set of $\binom{12}{2}$ candidates $C = \{c_{i,j} : i, j \in N, i < j\}$, one candidate for each pair of voters. First, assume that the chosen committee is $\{c_{1,2}, c_{3,4}, c_{5,6}, c_{7,8}, c_{9,10}, c_{11,12}\}$ (or equivalently any other committee covering all voters). Now let the first deleted candidate be $c_{1,2}$. Let $c'$ be the replacement candidate. We distinguish two cases. Firstly, if $c'$ is not approved by 1 or 2, we can assume without loss of generality that it is $c_{6,7}$. Then after deleting candidate $c_{3,4}$, in order to restore PJR+ one would need to cover at least three of $1, 2, 3, 4$ as otherwise one of $1, 2$ together with one of $3, 4$ would witness a PJR+ violation. This is impossible with just a single candidate. Therefore, $c'$ has to be approved by either 1 or 2. Without loss of generality, we can assume that it is $c_{2,3}$.

Next, we delete $c_{5,6}$. In order for $c_{1,5}$ or $c_{1,6}$ to not witness a PJR+ violation, we need to cover voter 1. If we include neither $c_{1,5}$ nor $c_{1,6}$, this leaves 5 and 6 uncovered. Therefore, after deleting for instance $c_{9,10}$ in the last step, we would again need to cover at least 3 of the uncovered voters, which is impossible with a single candidate. Hence, without loss of generality, we added $c_{1,5}$. We will immediately delete this candidate again, leaving all three of $1, 5, 6$ uncovered. In order, for 1 and 6 not to witness a PJR+ violation, we need to cover at least one of them. If we add $c_{1,6}$ we immediately delete it again. Now dependent on which candidate we add, we can either delete $c_{7,8}$, $c_{9,10}$, or $c_{11,12}$ and again get four uncovered voters, witnessing a PJR+ violation, which cannot be repaired with a single candidate. If we did not add $c_{1,6}$ two of $1, 5, 6$ are again uncovered and dependent on which candidate we added deleting $c_{7,8}$, $c_{9,10}$, or $c_{11,12}$ gets us four uncovered voters leading to a PJR+ violation.

Similarly, if we started with a committee leaving one voter uncovered, for instance $\{c_{1,2}, c_{3,4}, c_{5,6}, c_{7,8}, c_{9,10}, c_{10,11}\}$, after deleting $c_{1,2}$ we need to cover voter 12. If we do it with $c_{1,12}$, we can delete this candidate again immediately. If we then add $c_{2,12}$ and delete it immediately, we are left with four uncovered voters in any case, thus witnessing a PJR+ violation. If we do not add $c_{2,12}$ both $c_{1,2}$ are uncovered leading to a PJR+ violation after deleting $c_{3,4}$. Similarly, if we did not add $c_{1,12}$ in the first place, the voters 1 and 2 are uncovered. Thus, after deleting an appropriate next candidate, PJR+ cannot be restored. $\square$

**Theorem 5.4.** *There exists a robust fully-dynamic PJR+ algorithm making amortized 2 changes per iteration.*

*Proof.* Our goal is to maintain a maximally affordable subcommittee throughout the process. Let $C_0$ be the initial candidate set. We compute an affordable subcommittee $M_0$ of maximum size for $C_0$ and fill the remaining seats arbitrarily to initialize some committee $W_0 = M_0 \cup D_0$ with $D_0$ consisting of discardable candidates and initialize $k$ tokens.

For $t \geq 0$, at the start of the next time step $t + 1$, we add two tokens. When some candidate $c \in C_t$ is deleted from $C_{t+1}$, there are three cases. If $c \notin W_t$, we set $D_{t+1} = D_t$ and $M_{t+1} = M_t$; if $c \in D_t$, we refill $D_{t+1}$ by replacing $c$ with an arbitrary candidate $d \in C_{t+1} \setminus W_t$ and remove a token; and if $c \in M_t$, we first refund the budgets spent on $c$. Then, iteratively, we refill $M_{t+1}$ starting from $M_t$ by adding affordable candidates and dispose any candidate from $D_t$, as well as one token, until no candidate is affordable. If we do not add any candidate this way, we instead add $d \notin W_t$ to $D_t$ to obtain $D_{t+1}$, remove a token, and set $M_{t+1} = M_t \setminus \{c\}$. Whenever some candidate $c \notin C_t$ gets added to $C_{t+1}$, we check whether this candidate is affordable. If it is, we add it to $W'_t$ to obtain $M_{t+1}$ and remove any $d \in W_t \setminus M_t \neq \emptyset$, as well as a token.

Clearly, $M_t$ is maximally affordable and hence $W_t$ satisfies PJR+ in each time step $t$. To see that we need at most 2 changes per round amortized, note that in each round we add two 2 tokens. Hence, it suffices to show that the token count is always non-negative. Let $k' = |M_0|$ be the size of the initial committee and $t'$ be the number of *added* candidates. We claim that at each time-step $t$, there are at least $k' + t' - |M_t|$ tokens left. At time-step 0 this is clearly true, as we have not added any candidates or tokens yet. Thus, assume that it holds at time-step $t$. If we add a candidate $c_{t+1}$, we increase the number of tokens by at least one. Thus, the invariant still holds. If we delete a candidate not in the committee, the invariant is also still true, as we have just added two tokens. Similarly, if we delete from $D_t$ we add two tokens and remove one. Thus, finally, if we delete a candidate from $D_t$ we add new candidates until we reach a final maximally affordable subcommittee $M_{t+1}$. As the size of the original maximally affordable subcommittee was $k'$ and as we just added $t'$ candidates, we know that $D_{t+1} \leq k' + t'$. Therefore, the invariant still holds, even after deleting a candidate from $D_t$.

$\square$

**Theorem 5.5.** *There exists a fully-dynamic $\Theta(\log(k))$-EJR+ algorithm making amortized two changes per iteration.*

*Proof.* Let $H(n)$ denote the $n$-th harmonic number. Consider for the start a modified GJCR that considers groups of size $\geq H(k)\ell\frac{n}{k}$ instead of $\geq \ell\frac{n}{k}$. Again, this rule can be modeled via a budget of $\frac{k}{n}$ for each voter, which the voters then all uniformly spend when they can buy a candidate, which all have a unit cost. To show that on $C_0$ GJCR computes at most $k$ candidates, observe that each voter can buy at most 1 candidate for violations with $\ell = 1$, 2 for violations with $\ell \leq 2$ and so on. For a violation w.r.t. $\ell$, the price the voter pays it at most $\frac{k}{n\ell H(k)}$. In total, this yields $\sum_{\ell \leq k} \frac{k}{n\ell H(k)} = \frac{k}{n}$. So, no voter overspends, and since the

total budget was $k$, we have a committee of size $\leq k$. We fill up the remaining places with placeholders. Now, if a candidate is added and creates a violation of $H(k)$-EJR+ w.r.t. some $\ell \leq k$, then each voter that is part of this violation can have spent at most $\sum_{j \leq \ell} \frac{k}{njH(k)}$. To buy this candidate into the committee, they spend at most $\leq \frac{k}{n\ell H(k)}$ and thus no one overdraws their budget. Especially, since the budget was not fully used before, there are placeholders in the committee. Replace one of them with the newly added candidate, then $H(k)$-EJR+ is restored. Conversely, if a candidate $c$ is deleted and this creates violations of $H(k)$-EJR+, we can reimburse the voters who previously bought the candidate into the committee. With the same argument as in the instantiation and the addition of a candidate, we obtain that each voter has sufficient budget to buy the candidates causing the violations into the committee. There can be multiple of these changes to the committee after a single deletion. However, note that for $\ell \leq k$ changes to be made, there must have been $\ell$ total budget that was not used or freed beforehand. If the budget was freed, at most $\ell$ corresponding dummy candidates have been added to $W$ before. Together, these are at most $2\ell$ changes, preceeded by $\ell$ rounds in which the budget was freed or not used. $\square$

**Theorem 5.6.** *For any $\alpha > 1$ there exists an incremental $\alpha$-EJR+ algorithm making amortized $\frac{\alpha}{\alpha-1}$ changes.*

*Proof.* Let $\alpha > 1$ be given. For small $k$ with $k \leq \frac{\alpha}{\alpha-1}$, we can replace the entire committee in each step and trivially obtain the result. Else, we have $k > \frac{\alpha}{\alpha-1}$, implying $k(1 - \alpha) < -\alpha$ and thus $\frac{k}{\alpha} < k - 1$, which finally leads to the desired $\lceil \frac{k}{\alpha} \rceil < k$. Begin by running any EJR+ rule for the committee size $\lceil \frac{k}{\alpha} \rceil$ and select $k - \lceil \frac{k}{\alpha} \rceil$ other candidates arbitrarily. For the next $k - \lceil \frac{k}{\alpha} \rceil$ steps, while there is a candidate arriving who witnesses an $\alpha$-EJR+ violation, include them in exchange for one of the arbitrarily added candidates. Afterwards, recompute a committee of size $\lceil \frac{k}{\alpha} \rceil$ satisfying EJR+ and add them into the committee by excluding arbitrary candidates. After $k - \lceil \frac{k}{\alpha} \rceil + 1$ steps, we thus have made at most $k$ replacements. This leads to amortized $\frac{k}{k - \lceil \frac{k}{\alpha} \rceil + 1} = \frac{k}{k - (\lceil \frac{k}{\alpha} \rceil - 1)} < \frac{k}{k - \frac{k}{\alpha}} = \frac{1}{1 - (\frac{1}{\alpha})} = \frac{\alpha}{\alpha-1}$, replacements per round which concludes the proof.

$\square$

**Theorem 5.7.** *There exists an incremental EJR+ algorithm that is robust with respect to a single addition.*

*Proof.* To show this theorem, we use Algorithm 2. In essence, Algorithm 2 runs the GJCR with an additional local swapping step at the end. This local swapping step tries to maximize the number of voters covered in each iteration. As Algorithm 2 produces one possible outcome of GJCR, it satisfies EJR+.

**Algorithm 2** Locally Stable GJCR

---

$W \leftarrow \emptyset$ $N_{active} \leftarrow \emptyset$ **for** $\ell$ *in* $k, \ldots, 1$ **do**

   $W_\ell = \emptyset$ **do**

      **while** *there is* $c \notin W$*:* $|\{i \in N_c : |A_i \cap W| < \ell\}| \geq$ $\frac{\ell n}{k}$ **do**

         Choose $c$ maximizing $|\{i \in N_c : |A_i \cap W| < \ell\} \setminus N_{active}|$ $W_\ell \leftarrow W_\ell \cup \{c\}$ $W \leftarrow W \cup \{c\}$ $N_{active} \leftarrow N_{active} \cup \{i \in N_c : |A_i \cap W| < \ell\}$.

      **end**

   **while** *there was a change in the last iteration*;

   **for** $c \in W_\ell, c' \notin W$ **do**

      **if** $|\{\{i \in N_{c'} : |A_i \cap W \setminus \{c\}| < \ell\}| \geq \frac{\ell n}{k}$ *and* $|i \in N_{c'} : |A_i \cap W \setminus \{c\}| = 0\}| > |i \in N_c : |\{A_i \cap W \setminus \{c\}| = 0\}|$ **then**

         $W_\ell \leftarrow W_\ell \cup \{c'\} \setminus \{c\}$ $W \leftarrow W \cup \{c'\} \setminus \{c\}$ $N_{active} \leftarrow \{i \in N : |A_i \cap W| > 0\}$

      **end**

   **end**

**end**

return $W$

---

If Algorithm 2 outputs less than $k$ candidates, the theorem follows, as we can simply include the new candidate in the committee with a single swap, swapping out an irrelevant candidate. Thus, assume it outputs $k$ candidates and let $c$ be the newly added candidate witnessing an EJR+ violation. Let $N' \subseteq N_c$ be the set of voters witnessing the violation with $|N'| \geq \ell \frac{n}{k}$ and $|A_i \cap W| < \ell$ for all $i \in N'$. If $|A_i \cap W| = 0$ it is easy to see that the committee could not have been of size $k$. Therefore, every voter in $N'$ approves at least one candidate. Let $i \in N'$ be any such voter and let $\ell_i = |A_i \cap W|$. Let $c'$ be any arbitrary candidate in $A_i \cap W$ and consider the committee $W' := W \setminus \{c'\} \cup \{c\}$. Further, assume that $W'$ does not satisfy EJR+ with its violation being witnessed by candidate $c''$ for threshold $\ell''$ and set $N'' \subseteq N_{c''}$. We distinguish two cases:

**Case 1:** $\ell'' > \ell_i$. Then, in iteration $\ell''$ some voter in $N''$ must approve at least $\ell''$ candidates, one of which must be $c'$. Therefore, $c'$ got bought in an iteration before $\ell_i$ a contradiction.restatable

**Case 2:** $\ell'' < \ell_i$. Then, someone in $N''$ must approve of $c'$. Since $\ell'' < \ell_i$ this voter must approve less than $\ell_i$ candidates in iteration $\ell_i$ and must therefore have contributed to buying $\ell_i$. Thus, this voter approves at least $\ell_i - 1$ candidates in $W'$ contradicting $\ell'' < \ell_i$.

**Case 3:** $\ell'' = \ell_i$. Let $N''_1 = \{j \in N'' : c' \in A_j\}$. If $N''_1 = N''$ this set must necessarily include $i$ who still approves $\ell''$ candidates in the outcome, a contradiction. Therefore, there is a $j \in N'' \setminus N''_1$. However, since for GJCR to select $k$ candidates, every "buyer" of $c'$ must approve exactly $\ell_i$ candidates. Otherwise, in the price-system constructed by

GJCR, one of these "buyers" must pay less than $\frac{k}{n}$, leading to a contradiction that we selected $k$ candidates (see Brill and Peters (2023, Proposition 8) for a full proof). Therefore, we could have swapped $c''$ with $c'$ increasing the number of covered voters in iteration $\ell_i$ by at least 1, as there is no voter who go down to 0 approvals after the removal of $c'$. (The implicit assumption here is that $\ell_i$ is at least 2, which must be true, as otherwise $j$ approves nothing, and we would not have selected $k$ candidates.)

$\square$

