# OpenReview forum: "Maintaining Proportional Committees with Dynamic Candidate Sets"
_ICML.cc/2025/Conference — ICML 2025 poster_

### Official Review · Reviewer_hSUk · 2025-03-08

**Overall Recommendation:** 3

**Summary:**

This paper consider a series of problems about modifying the winner set of a multi-winner election when there are changes on candidates. The paper investigate three types of voter preferences: rankings, metric-space, and approval (0-1). Both positive (algorithms satisfying certain fairness axioms) and negative (some fairness axioms cannot even be approximated) theoretical results are given.

**Claims And Evidence:**

Yes, the claims are supported by the Theorems and proofs.

**Essential References Not Discussed:**

No.

**Experimental Designs Or Analyses:**

This paper have no experiments.

**Methods And Evaluation Criteria:**

Yes (once the proofs are correct).

**Other Comments Or Suggestions:**

No.

**Other Strengths And Weaknesses:**

Strength: The paper studies an interesting and well-motivated extension of multi-winner voting. The authors study the problem under three different preferences models and prove a bunch of non-trivial theoretical results, giving a somewhat thorough characterization. The paper also well placed itself in the previous literature.

Weaknesses: There can be a large improvements on the presentation of the paper, especially the proof. The proofs lack intuition, intention, and explanations on each step, making it non-trivial to understand what each step is doing. This applies to the contradiction method in Theorem 3.1 (what makes a contradiction? specify it) and almost everything in Theorem 4.1. The paper will largely benefit from adding intuitive proof sketch, running examples, and more discussions.

For now I will give a negative score. If the authors could persuade me that their Theorem 4.1 is correct, I am happy to raise my score.

------
Score updated.

**Questions For Authors:**

1. Explain you proof for Theorem 4.1 and the contradiction for Theorem 3.1. Address my questions in the ''correctness' part.

**Relation To Broader Scientific Literature:**

This paper expands the discovery of multi-winner election to the scenario where the candidates may change. The potential significance of the problem is illustrated in the introduction of this paper, where a series of scenarios with changeable candidates are given.

**Theoretical Claims:**

I checked Theorem 3.1 and Theorem 4.1

I think Theorem 3.1 is correct.

For Theorem 4.1, I would be careful to draw the same conclusion. The proof is extremely confusing as many steps are unexplained. The proof starts with creating clusters, while many details are omitted (such as what are the $p_j(i)$, and what if two clusters with same diameters comes together). Then it goes into show the envy relations keeps invariant in the modifications, but I don't see this is applied anywhere in the proof. Finally, the real part for the axioms seems have nothing to do with the dynamic changes. For Right Column of Line 277-286, the formula, \rho$ is undefined. I doubt it is the approximation ratio Gamma for proportionally fairness.

--------------

After rebuttal: theorem 4.1 is further explained and looks better. I encourage the authors to improve the presentation of the proof.

---

> ### Author Rebuttal · Authors · 2025-03-31
>
> - Explain your contradiction for Theorem 3.1
>
> Answer: We are given a profile with $n$ voters and a committee $f(t-1)$ that previously satisfied PSC. At time $t$, a new candidate $c_t$ becomes feasible, and $f(t-1)$ may now fail to satisfy PSC -- we would like to fix that with a single swap involving $c_t$. Assume that we cannot do so, then $f(t-1)\cup \{c_t\}\setminus \{b\}$ violates PSC for all $b\in f(t-1)$.
> In total, this yields $k+1$ committees that violate PSC. For each of these committees, we prove that we can reserve $\frac nk$ voters. Thus, in total there are $(k+1)\frac nk$ voters -- more than $n$, a contradiction.
>
> - Explain your proof for Theorem 4.1.
>
> Answer: We apologize for the confusion created here, we agree that we were a bit sloppy in writing the proof. We are sorry for this.
> On a high level our proof works as follows: we start off with a pre-clustering phase, leading to $k$ groups of agents $N_1$ to $N_k$. In particular, in the way we define it an agent can be part of multiple groups. To construct these groups, we use a common technique from other papers, namely, we give each of the $n$ agents a budget of $k/n$. With $k$ groups to be opened at cost of $1$ each. this intuitively means that each agent can open their "proportional share" of a group. We now open these groups by letting the agents buy groups using their share of the budget, and we assign each agent to a group they paid for. Now, after creating these groups, we go one-by-one and let them pick the cluster center that is closest to them. When a point is added, we update these picked cluster centers and potentially swap cluster centers between these groups (that is the invariant envy relation). In the end, we use this fact that every group has their "favourite" cluster center to bound the factor by which any set of n/k agents can improve.
>
> - The proof is extremely confusing as many steps are unexplained. The proof starts with creating clusters, while many details are omitted (such as what are the p_j(i)
>
> Answer: In general, we will make sure to mark clearly the properties that are important for the high level idea of the proof, and state which arguments are only required to prove these important properties.
> E.g., it does not really matter how the prices are chosen and we agree that we should explain that. To show that it is possible to define prices, one way to do so is: (1) check for the first voter in $N'$, i.e., $i_1 = \min\{i\in N': b_i > 0\}$ whether she has a budget of $1$ on her own. If yes, we reduce her budget by $1$ and only let her pay for the cluster. (2) If no, voter $1$ pays all she has and we set her remaining budget to zero. Consider the next voter in $N'$ with positive budget. If $i_2$ can pay for the remaining cost, she does so and we reduce her budget accordingly. (3) Otherwise, she contributes all of her remaining budget, we set her budget to zero and continue.
>
> - and what if two clusters with same diameters comes together).
>
> Answer: If there are ties any of them can be chosen, i.e., we assume some arbitrary tie-breaking. We will note this in the paper.
>
> - Then it goes into show the envy relations keeps invariant in the modifications, but I don't see this is applied anywhere in the proof.
>
> Answer: The envy relation is just technical and guarantees us something important: the chosen committee contains the closest candidate to each cluster $N_i$. We apply this in line 320. Here, $i$ belongs to a cluster $N''$, and hence we can assume that cluster member is closer to the chosen committee than i is to c.
> We will state our goals more clearly in the proof.
>
> - Finally, the real part for the axioms seems have nothing to do with the dynamic changes.
>
> Answer: Indeed, the guarantee holds at every time step. Note that the tricky, dynamic part of the algorithm is described within the envy cycle procedure -- here, when candidates join or leave the election, we determine, depending on how the envy of the clusters, whether we need to modify the committee for this time step.
>
> - For Right Column of Line 277-286, the formula, rho is undefined. I doubt it is the approximation ratio Gamma for proportionally fairness.
>
> Answer: Sorry about this, indeed rho is supposed to be gamma, i.e., the improvement factor or approximation ratio. We will declare more clearly where the variable comes from.

---

> > ### Comment · Reviewer_hSUk · 2025-04-03
> >
> > Thank you for your explanation. Though I am not totally confident in understanding the proof, your high-level explanation helps. I will raise my score.

---

### Official Review · Reviewer_gYxQ · 2025-03-12

**Overall Recommendation:** 5

**Summary:**

This paper introduces a temporal element to multi-winner voting by studying a model wherein the set of candidates changes over time. This is separated into three settings: incremental (candidates are added over time), decremental (candidates are removed over time), and fully dynamic (candidates can be added or removed over time). These settings are each studied under three different paradigms of multi-winner voting: ranking-based, approval-based, and clustering.

Within each paradigm two proportionality axioms are studied. The paper shows that, across paradigms, proportionality can typically be achieved or approximated when candidates are added over time. When candidates are removed or in the fully dynamic setting proportionality can sometimes be approximated.

**Claims And Evidence:**

The claims in the paper are described quite clearly. Some theorems are supported by a proof in the main text (while other proofs are relegated to an appendix) and the claims all seem to be quite reasonable and attached to some evidence.

**Essential References Not Discussed:**

N/A

**Experimental Designs Or Analyses:**

N/A

**Methods And Evaluation Criteria:**

The paper is entirely theoretical and conceptual. The proofs that I considered were well-written and are the suitable method for supporting a theorem.

**Other Comments Or Suggestions:**

Tiny issues to adjust:

- missing a space on line 3 of section 3
- line 336 should probably include W: "a committee $W$ satisfies"

**Other Strengths And Weaknesses:**

In general I find the paper to be quite well written. Put bluntly, I typically find purely theoretical papers to be quite a slog and this was much easier to read than I had anticipated. In considering three paradigms of multi-winner voting and three types of dynamic candidates the paper manages to quite naturally fit in an impressive amount of content while remaining fairly readable.

Many temporal settings have been studied in the single winner setting (e.g. in iterative voting settings) but I have not previously seen work on this dynamic candidates in multi-winner settings. The motivation is quite natural and makes understanding the task seem a useful step towards approaching real-world applications.

Personally, I would consider the paper to be more useful in a setting where page limits are not an issue. This might be better as a journal paper but I recognize that in our current time a conference is the expected venue for publication.

Relatedly: while the paper fits in quite a bit already and is focused on establishing early theoretical results I can easily imagine some experiments being informative. In particular, I would be interested in understanding how often the various axioms you study are violated by various voting rules under different parameters (e.g. number of voters, number of candidates, preference distributions).

**Questions For Authors:**

I am unlikely to update my evaluation based on your response. Feel free to respond, or not, to any portion of my review. I would be interested to hear your perspective on the informativeness of any experiments that might be done on this setting.

**Relation To Broader Scientific Literature:**

The paper builds upon three existing paradigms of multi-winner voting and a specific priority (proportionality) which is popular within the multi-winner setting. Much work has been done on these topics; this paper adds what seems to be a fairly novel -- but quite natural -- component of dynamicity. The paper fits quite neatly into contemporary work on multi-winner elections within computational social choice.

**Theoretical Claims:**

I lightly reviewed the proofs that are included in the main text and found no issues.

---

> ### Author Rebuttal · Authors · 2025-03-31
>
> Thank you very much for the kind review!
>
> - I would be interested to hear your perspective on the informativeness of any experiments that might be done on this setting.
>
> Answer:
> We have thought about this for a bit. There are a few possible experiments we could see. Firstly, we already did a bit of experimentation using preference sampling, namely, we tried to find simple examples akin to Observation 1 or Example 3 in sampled data (e.g., we used some preference distributions for approval preferences, ran common voting rules such as MES or PAV and tried to see if the committees selected by them are not robust to a single deletion). However, we were not able to find any such violations in the short experiments we conducted.
>
> Secondly, what might be a bit more interesting would be to experiment on real-world data. Namely, there are two recently "uncovered" large datasets from last year, one being a dataset of approval-based multiwinner voting elections for so-called proof-of-stake elections by Boehmer et al. [1] and the other one being a dataset of over 1000 multiwinner ordinal preference elections from local Scottish government elections by McCune and Graham-Squire [2]. In particular, we would find it quite interesting to evaluate the effect of so-called "by-elections", i.e., elections to replace a candidate who dropped out of the parliament. In particular, these by-elections are not done using a proportional voting method, but by employing the singlewinner instant runoff voting rule. It would be interesting to see, if somehow one can measure the impact of this on the proportionality of the outcome.
>
> Interestingly enough, the data provided by Boehmer et al., is actually temporal. Namely, in these blockchain elections, there is one election per day. We would find it interesting to see, if one can perhaps use this data to evaluate our findings and to actually check whether dynamic multiwinner voting is hard. (We note that their data comes with the added difficulty that it allows voters to change their preferences)
> References:
>
> [1] Boehmer, Brill, Cevallos, Gehrlein, Sánchez-Fernández, Schmidt-Kraepelin, 2024, AAAI, Approval-based committee voting in practice: a case study of (over-) representation in the Polkadot blockchain
>
> [2] McCune and Graham-Squire, 2024, Social Choice and Welfare, Monotonicity anomalies in Scottish local government elections

---

> > ### Comment · Reviewer_gYxQ · 2025-04-02
> >
> > Thank you for the response. This empirical evaluation you discuss sounds like it would be quite interesting.

---

### Official Review · Reviewer_ut2U · 2025-03-13

**Overall Recommendation:** 4

**Summary:**

The paper considers multiwinner voting rules when the candidate sets are dynamic in one of 3 ways - candidates arrive one at a time, leave one at a time, or a mix of both. They consider 3 different types of preference classes - ordinal, distance-based, and approval based. They show that in some settings, there exist voting rules that achieve proportionality (which is defined differently depending for each preference class) whereas in others, there cannot exist any such rule.

**Claims And Evidence:**

Yes, the claims are appropriately cited or proven.

**Essential References Not Discussed:**

N/A

**Experimental Designs Or Analyses:**

N/A

**Methods And Evaluation Criteria:**

N/A

**Other Comments Or Suggestions:**

Page 4, Near Line 184 ".As there are .." missing a space after the period.

Page 5, proof of Theorem 4.1, I got briefly confused because $i$ is used for both the voters and the index of the clusters $N_i$ which have no relation to the individual voters. I would suggest a different indexing variable.

Page 13, Proof of Proposition 3.2. It should say $t<s$, not $s<t$.

Page 14, Proof of Theorem 5.3, the target size should be $k=6$, not $k=4$.

**Other Strengths And Weaknesses:**

The online models they consider are reasonable, and their results in these models appear promising. They also study several preference models. I also particularly like all the negative results they have, especially for existing voting rules.

My only (minor) complaint would be that the proofs don't seem very technically novel, or at least it is unclear to me what the novel parts are. If there are any new ideas, it would be better if the authors could highlight them more.

**Questions For Authors:**

N/A

**Relation To Broader Scientific Literature:**

There appear to be several related works in related online settings, but none in their specific models.

**Theoretical Claims:**

As far as I can tell, the proofs in the main body of the paper are correct.

---

> ### Author Rebuttal · Authors · 2025-03-31
>
> Thank you for the comments and suggestions. We implemented the changes (for the next version).

---

### Decision · Program_Chairs · 2025-05-01

**Decision:**

Accept (poster)

**Comment:**

In the end, all the reviewers viewed the paper positively. The idea of dynamically changing the candidate set is novel and is certainly worth presenting. The paper is well written, especially given its theoretical nature.